# Study on the Influence of Deformation and Temperature on the Properties of High-Strength Tungsten Alloy Wire

**DOI:** 10.3390/mi16080922

**Published:** 2025-08-10

**Authors:** Junling Fan, Jingwen Du, Jun Cao, Yongzhen Sun, Junchao Zhang

**Affiliations:** 1School of Chemical and Environmental Engineering, Jiaozuo University, Jiaozuo 454000, China; 2School of Mechanical and Power Engineering, Henan Polytechnic University, Jiaozuo 454000, China

**Keywords:** W-La alloy, microstructure, mechanical properties, fracture morphology

## Abstract

In this paper, high-strength W-1%La_2_O_3_ alloy wire was obtained by solid-state doping using tungsten powder and lanthanum oxide, large deformation rotary forging and wire drawing, which solved the disadvantages of traditional tungsten alloy wire processing such as the uneven distribution of rare earth oxides. The effects of rotary forging and annealing on the microstructure and properties of tungsten alloy were studied, which provided some basis for preparing high-strength tungsten alloy wire. The results indicate that tungsten alloy undergoes recovery at relative high temperatures (1480–1380 °C) during the rotary forging process. After large deformation, subgrains and uneven microstructures appear, so annealing is required before tungsten alloys wire drawing processing. With increasing annealing temperature, the recrystallization degree gradually increases and the hardness of tungsten alloy gradually decreases. When the deformation is less than 81.2%, tungsten alloy wire exhibits brittle fracture. When the deformation increases to 88.4% (ø0.8 mm), the fracture surface of the wire exhibits a plastic–brittle mixed fracture mechanism.

## 1. Introduction

Tungsten wire is divided into pure tungsten wire [1], doped tungsten wire [2] and tungsten alloy wire. Owing to its high melting point and high-strength at elevated temperatures, tungsten wire is widely used as filaments, heating elements, thermocouples and cathode materials in electron tubes [3]. In recent years, high-strength tungsten wire has been employed as the core wire for diamond wire saws in the field of photovoltaic silicon wafer cutting [4]. Tungsten wire is gradually replacing high carbon steel wire because of its high strength, good wear resistance and good machinability [5]. However, the inherent brittleness of tungsten imposes limitations on its processing and applications. Numerous studies have found that the mechanical properties of tungsten alloys can be improved by adding various rare earth oxides, rhenium, potassium and other elements [6]. Therefore, developing advanced manufacturing techniques for tungsten alloy wire remains an urgent and challenging task.

Yang Yongbin [7] found that the microstructure distribution of tungsten alloy after rotary forging was uneven, with axial grains being elongated and the outer layer exhibiting greater deformation than the central region. After heat treatment, the tensile strength of the alloy decreased slightly, while the elongation increased, significantly enhancing the material’s ductility and thereby facilitating subsequent plastic processing. N. Kaan Calıskan [8] reported that rotary forging enhanced the relative density of 90W–7Ni–3Fe alloy, along with its elongation and tensile strength. R. Osama [9] studied the effect of 10–50% deformation on 0.03% Y_2_O_3_ W-Ni-Fe alloy by rotary forging of the alloy in different degrees. The results demonstrated that increasing deformation led to a gradual rise in hardness and tensile strength, but a reduction in elongation. Zhang Linhai [10] observed that with increasing rotary forging deformation, the density and hardness of tungsten rod initially increased rapidly, then rose more slowly, eventually plateauing. Significant material deformation heterogeneity is prone to occur during the rotary forging of tungsten rods. Consequently, annealing is typically employed to enhance the material’s wire drawability [11].

Yu Ming [12] and Thomas Larsen [13] reported that tungsten recrystallized during annealing, with the hardness of pure tungsten plate gradually decreasing with the prolonged of annealing time. Xiang Zan et al. [14] prepared Y_2_O_3_ tungsten plates via powder metallurgy. Their findings indicated that, compared with pure tungsten, the Y_2_O_3_ addition delayed the recrystallization process of tungsten plates. Zhang, T., et al. [15] prepared a W-1 wt%Re-0.5 wt%ZrC alloy and observed that adding Re in combination with ZrC particles significantly increased the recrystallization temperature of tungsten alloy to between 1600 and 1700 °C. Vladica Nikolić et al. [16] studied the effect of annealing in the range of 900 °C to 1600 °C on pure tungsten wire and K-doped tungsten wire. Their research demonstrated that pure tungsten wire was recrystallized between 1300 °C and 1500 °C, resulting in the disappearance of the fibrous grain structure and the formation of coarse grains. In contrast, the K-doped tungsten wire maintained a fibrous grain structure within this temperature range, indicating a recrystallization temperature exceeding 1600 °C.

Previous studies have analyzed the microstructure and properties of tungsten materials processed by rotary forging and annealing, yet the evolution of microstructure and brittle toughness transition mechanism during sequential processing remains unclear. In this study, ø0.8 mm W-1%La_2_O_3_ alloy wire was fabricated via plastic forming (rotary forging and wire drawing). The effects of rotary forging process parameters, annealing temperature and drawing deformation on the microstructure and properties of tungsten alloys were studied, revealing the brittle-to-ductile transition mechanism of lanthanum-doped tungsten alloys and providing a theoretical basis for the preparation of high-strength lanthanum-doped tungsten alloy wires.

## 2. Materials and Methods

The material used in this study was W-1%La_2_O_3_ alloy. Starting with APT (Ammonium Paratungstate) as the raw material, blue tungsten oxide (BTO) was produced via hydrogen reduction. La_2_O_3_ was then added and blended with the BTO. Subsequently, the mixture was reduced in a 15-tube reduction furnace to obtain alloyed tungsten powder with a particle size distribution of 2.0–2.5 μm. The powder was then sieved through a 200-mesh screen to remove coarse particles and impurities. The sieved powder was compacted by cold isostatic pressing at 200 MPa for 90 s, followed by sintering in a medium-frequency furnace for 23 h (the heating profile is shown in Figure 1). This process yielded sintered tungsten alloy billets with a diameter of ø23 mm. The tungsten alloy billet was then cogged on a rolling mill. The billet was heated to 1650 °C and held for 30 min prior to rolling. After ten passes, the ø8.9 mm rod was obtained. Subsequently, the tungsten alloy rods were annealed in a hydrogen atmosphere using a high-frequency induction furnace. The annealing temperature ranged from 2000 °C to 2300 °C with a holding time of 10 s. After that, the ø8.9 mm rod underwent two cycles of deformation and annealing to achieve a final diameter of ø5.2 mm. Finally, the ø5.2 mm rod was processed into ø0.8 mm tungsten alloy wire through a series of wire drawing passes.

During the plastic forming of tungsten rods, the degree of deformation of the material is expressed by deformation. The data can be expressed by the following formula [17]:*ε* = (*D*_2_ − *d*_2_)/*D*_2_*ε*—deformation, %; *D*—diameter of material before deformation; *d*—diameter of material after deformation.

The experimental design for the rotary forging process is detailed in Table 1. Two sets of annealing temperature tests are summarized in Table 2. Annealing was performed using an intermediate frequency induction furnace equipped with a 16 cm diameter induction coil. The wire feed speed through the annealing zone was 0.93 m/min, resulting in an annealing time of approximately 10 s. Finally, wire drawing experiments, as outlined in Table 3, were conducted to fabricate ø0.8 mm tungsten alloy wire.

The tungsten alloy specimens were sectioned using an HF320M CNC EDM wire-cutting machine (Anhui Zhongke Chungu Laser Industrial Technology Research Institute Co., Ltd. Anhui, China). Subsequently, they were ground and polished on a MoPao160 double-speed grinder/polisher. Metallographic etching was performed using a mixed solution of hydrogen H_2_O_2_ and ammonia, in a volume ratio of 3:1. FX-41MW trinocular inverted metallographic microscope(Changsha Miqi Instrument Equipment Co., Ltd. Changsha, China) and JSM-6390LA(Japan Electron Optics Laboratory Co., Ltd. Tokyo, Japan) and Merlin Compact(Carl Zeiss AG Co., Ltd., Oberkohen, Germany) SEM (scanning electron microscopes) were used for the observation of microstructures and the analysis of the EDS (Energy Dispersive Spectrometer)((Carl Zeiss AG Co., Ltd., Oberkohen, Germany)). The tungsten alloy samples were investigated by EBSD (electron back scatter diffraction) to study the orientation distribution and weave structure. The Vickers hardness (HV, MPa) of tungsten alloy rod was measured with a TH702 digital microhardness tester(Beijing TIME High Technology Ltd. Beijing, China) under a test load of 1000 gf and a dwell time of 10 s. Three measurements were taken on each sample surface and use the average value to reduce the error. The density of the samples was measured by Archimedes method, and the theoretical density of tungsten 19.35 g/cm^3^ and the theoretical density of lanthanum oxide 6.51 g/cm^3^ were taken to calculate the relative density of tungsten alloy samples. 

## 3. Results and Discussion

Figure 2 presents the grain boundary maps and the grain orientation distribution of tungsten alloys during different processing states. The high fractions of low-angle grain boundaries (LAGBs) in Figure 2a,c account for 48% and 49%, respectively, which indicates that the internal grains of the alloy are severely deformed and there is a high local dislocation density within the alloy following large deformation. This elevated dislocation density promotes stress concentration, increasing the material’s susceptibility to cracking. In contrast, Figure 2b shows a predominance of high-angle grain boundaries (HAGBs) at 87.5%, accompanied by significant grain growth. This suggests that recrystallization occurred during annealing. This suggests that after annealing treatment, crystal recrystallization occurs, the large angle grain boundaries increase the crystal energy and the original diffuse strengthening effect of the alloy is weakened or disappears, resulting in a reduction in strength [18].

It can be seen from Figure 2a that the tungsten alloy with 85% rolling deformation shows a relatively uniform orientation distribution without a pronounced surface texture [19]. After annealing, the grains evolve towards an equiaxed morphology with improved orientation uniformity, although some elongated grains persist. A strong <110> fiber texture develops in the final drawn ø0.8 mm tungsten alloy wire (Figure 2c), demonstrating that <110> becomes the dominant texture orientation along the wire axis as plastic deformation progresses [20].

### 3.1. The Influence of Rotary Forging on the Microstructure of Tungsten Alloy

Previous studies indicate that the particles in W-1%La_2_O_3_ tungsten alloy contain not only La_2_O_3_ but also tungsten oxide and a tungstate containing lanthanum [21]. Figure 3a displays the metallographic microstructure of the cross-section of the ø23 mm tungsten billet. The microstructure exhibits diverse grain morphologies, distinct grain boundaries and a dense structure. Figure 3b illustrates the grain size distribution across the cross-section of tungsten alloy. The grain sizes range from 5 μm to 50 μm, with the majority falling between 10 μm and 30 μm. The average grain size is 21.33 μm. The grain size distribution approximates a normal distribution, indicating a relatively uniform microstructure.

The examination of the microstructure of ø8.9 mm tungsten alloy rods annealed at 1480 °C revealed the presence of white spherical or elliptical particles on the W-La alloy matrix, as shown in Figure 4. To identify their composition, spot EDS analysis was performed on these particles; the results are shown in Table 4. The EDS data show that W and O are the predominant elements, while La content is minimal. This suggests that oxidation occurred during the preparation process, and the white particles are primarily tungsten oxides.

Dark spherical and acicular particles were also observed in the microstructure of the ø8.9 mm tungsten alloy rod annealed at 1480 °C, as shown in Figure 5. Spot EDS analysis was performed on these particles, and the results are shown in Table 5. As seen in Table 5, the content of La element at points 5, 6 and 7 is significantly higher, suggesting that La forms thermally stable La_2_O_3_ particles during alloy processing. These dark particles are thus identified as lanthanum oxide particles [22]. Figure 6b shows a generally uniform distribution of lanthanum within the tungsten alloy matrix, although localized clustering is evident. In Figure 6a, numerous dark irregular particles are dispersed both within the tungsten matrix and along the grain boundaries. These oxide particles inhibit grain growth, hinder dislocation motion, and contribute to strengthening via the Orowan mechanism [23,24,25]. Concurrently, grain refinement increases the grain boundary density, leading to significant fine-grain strengthening and an overall improvement in alloy properties [26].

Figure 7 presents the longitudinal section microstructure of tungsten alloy processed under different temperatures and deformation conditions. Comparing microstructures under identical deformation (85% rolling), the fibrous grain morphology in the longitudinal section appears qualitatively similar for both low- and high-annealing temperatures. Figure 7a,b illustrate the longitudinal section microstructure of tungsten alloy rod obtained after 85% rolling deformation. This large deformation results in a fragmented grain structure, with fine grains observed both within and along the boundaries of the original crystals. Quantitative analysis reveals distinct grain size variations with processing temperature under the same deformation: Figure 7c: Average long axis ≈ 95.1 μm, short axis ≈ 18.2 μm (aspect ratio ≈ 5.2); Figure 7d: Average long axis ≈ 78.8 μm, short axis ≈ 11.2 μm (aspect ratio ≈ 7.0); Figure 7e: Average long axis ≈ 154.6 μm, short axis ≈ 14.3 μm (aspect ratio ≈ 10.8); and Figure 7f: Average long axis ≈ 104.2 μm, short axis ≈ 7.8 μm (aspect ratio ≈ 13.4). Therefore, under identical deformation (85% rolling), increasing the processing temperature leads to a decrease in the average dimensions of both the long and short axes of the fibrous grains. However, concurrently, the grain aspect ratio (long axis/short axis) increases significantly. Furthermore, at a constant processing temperature, increasing the deformation level from 34.6% to 65.9% also results in an increase in the aspect ratio of the fibrous grains, making them more elongated. Figure 7g,h depict the microstructure of ø5.2 mm tungsten alloy rod processed at 1330 °C and 1380 °C, respectively, under a higher deformation level of 82.7%. The longitudinal microstructure at both temperatures exhibits elongated grains, but with an increased grain width compared to lower deformation levels. This significant microstructural change, characterized by grain coarsening, indicates the onset of recrystallization and grain growth.

Figure 8 presents the cross-section microstructure of tungsten alloy at different temperatures. Comparing Figure 8a and Figure 8b, cracks are observed in the sample processed at 1480 °C, which also exhibits a lower grain count compared to the sample processed at 1400 °C. This is attributed to excessive processing temperature, which increases the material’s susceptibility to high-temperature brittleness, facilitating crack initiation at grain boundaries. Figure 8c–f show the microscopic morphology and grain size distribution of tungsten alloy samples processed at different temperatures. The corresponding average grain sizes are 8.46 μm, 14.73 μm, 16.29 μm and 17.52 μm, respectively.

The average grain size at 1450 °C and 1400 °C are larger than those at 1380 °C and 1350 °C. This indicates that during rotary forging, two competing phenomena occur: grain refinement due to deformation and grain coarsening due to dynamic recovery/softening at elevated temperatures. The fragmented grains undergo recovery during processing. Furthermore, the grain size statistics in Figure 8c–f reveal that increasing deformation leads to a more uniform microstructure. Figure 8g,h depict the microstructure of the tungsten alloy samples after secondary annealing. At 1330 °C (Figure 8g), numerous subgrains are present. In contrast, at 1380 °C (Figure 8h), larger grains are observed. Recrystallization occurs at both temperatures, with a higher degree/extent observed at 1380 °C. Therefore, excessively high processing temperatures can induce recrystallization in tungsten alloys, exacerbating high-temperature brittleness. This promotes the formation of internal cracks and can even lead to fracture of tungsten alloy rods during subsequent processing. Thus, annealing after large deformation processing is essential before wire drawing. The recommended temperature range for rotary forging processing is 1330 °C to 1400 °C.

### 3.2. The Influence of Rotary Forging on the Hardness of Tungsten Alloy

The relationship between hardness and grain size of tungsten alloy can be expressed by the following formula [17]: *H = H*_0_
*+ K_H_·d*^−1/2^. (Formula: *H*—hardness of tungsten, MPa; *H*_0_—constant; *K_H_*—constant; *d*—average grain diameter.)

From the above formula, the hardness of the material increases gradually with the decrease in grain size. The W-La alloy obtained by adding La_2_O_3_ into pure tungsten to achieve a uniform distribution of oxide particles, exhibits a smaller average grain size and higher hardness than pure tungsten. The enhanced hardness of W-1%La_2_O_3_ compared to pure tungsten arises from the combined effects of dispersion strengthening and grain refinement [27]. As fine dispersed particles, particles are uniformly distributed within the tungsten matrix. These particles hinder dislocation movement via the Orowan mechanism, providing significant strengthening, thereby improving the alloy hardness.

Table 6 lists the hardness, density and relative density of all samples. As shown in Figure 9, the alloy rod processed at lower temperatures (1330 °C to 1400 °C) exhibits a hardness of 522.2 HV. This high hardness is attributed to the substantial work hardening resulting from the large deformation (reducing the diameter from ø23 mm to ø8.9 mm). After annealing during processing at a deformation level of 34.6%, the alloy hardness decreases to 463.8 HV. When the deformation increases from 34.6% to 65.9%, the hardness increases to 498.0 HV. This increase is caused by work hardening due to the elevated dislocation density induced by plastic deformation. Further increasing the deformation to 82.7% raises the hardness significantly to 605.2 HV. This substantial increase results from the combined effects of increased deformation and the associated pronounced work hardening.

At higher processing temperature (1480–1380 °C), the hardness of tungsten alloy will decrease. Elevated temperatures promote grain growth and grain boundary migration, leading to microstructural changes that reduce hardness. While increased deformation generally enhances dislocation density and refines grains, causing work hardening and thus increasing hardness, this effect is counteracted at high temperatures. When deformation increases from 34.6% to 82.7% at high temperatures, the hardness remains relatively constant. This is because dynamic recovery and grain growth at high temperatures offset the work hardening induced by large deformations.

As can be seen from Figure 10, the relative density of tungsten alloy is above 97% at both low temperature and high temperature, and the density of tungsten alloy gradually increases with the increase in deformation. When the deformation increases from 0% to 82.7% at low temperature (1400–1330 °C), the density increases from 18.42 g/cm^3^ to 18.76 g/cm^3^, and the relative density increases from 97% to 98.8%. At higher processing temperature (1480–1380 °C), the density increases from 18.48 g/cm^3^ to 18.81 g/cm^3^, and the relative density increased from 97.4% to 99.1%, indicating that the material achieves near-theoretical density. The higher density observed at elevated temperatures is attributed to enhanced sintering densification. Elevated temperatures promote grain growth and reduce porosity, particularly at grain boundaries, thereby increasing the material density [28].

### 3.3. The Effect of Heat Treatment on the Microstructure and Hardness of Tungsten Alloy

After annealing, the internal grain structure of tungsten alloy rod has been adjusted, enhancing its workability. The grain size of the annealed samples must be controlled to ensure smooth subsequent processing and the quality of the drawn wire. Table 7 presents the grain size and hardness of the tungsten alloy after annealing at different temperatures. Annealing was performed in the temperature range of 2000 °C to 2300 °C for approximately 10 s. The grain granularity of ø8.9 mm tungsten alloy is 1461, 1484 and 2296 grains/mm^2^ after annealing at different temperatures. Empirical data indicate that optimal subsequent processing quality for this size is achieved when the annealed grain density falls within the range of 1300–1500 grains/mm^2^. For the ø5.2 mm tungsten alloy rod, the grain density after annealing reached 1496, 619 and 2263 grains/mm^2^, respectively. According to experience, the annealed grain density for this size should be at least 1000–1200 grains/mm^2^ to ensure good processability. The low grain density (619 grains/mm^2^) observed in sample 2 annealed at 2250 °C may lead to quality issues during subsequent processing.

Figure 11 presents cross-sectional microstructure of the ø8.9 mm tungsten alloy rod annealed at different temperatures. After annealing at 2000 °C (Figure 11a), fine subgrains are observed at grain boundaries, indicating incomplete recrystallization as significant grain growth has not occurred. Figure 11b shows that grains gradually coarsen and the fraction of recrystallized grains increases. At an annealing temperature of 2300 °C (Figure 11c), further grain growth occurs, and fine subgrains evolve into new equiaxed grains.

Figure 12a reveals a lower recrystallization degree and finer grain size at an annealing temperature of 2250 °C. Upon increasing the temperature to 2250 °C, the recrystallization degree increases and the grains grow. Only a few subgrains are present within the grains and at the grain boundaries, as shown in Figure 12b. When the annealing temperature rises to 2300 °C, the grains grow further and new equiaxed grains form [29], as shown in Figure 12c. Therefore, it can be inferred that the recrystallization temperature of tungsten alloy is approximately 2300 °C.

The hardness change is shown in Figure 13. The hardness of the ø8.9 mm tungsten alloy is 466.5 HV after annealing at 2000 °C. At an annealing temperature of 2100 °C, the hardness decreases from 466.5 HV to 440.5 HV due to internal recovery and slight recrystallization of the tungsten alloy, resulting in changes in the microstructure. When the annealing temperature is 2300 °C, recrystallization is completed within the crystal and the grains grow slightly in size, eliminating the work hardening caused by high deformation processing, resulting in a decrease in hardness to 395.3 HV. As the temperature increases from 2200 °C to 2250 °C, the dislocation density decreases, while recrystallization and grain growth occur, resulting in a decrease in hardness from 435.2 HV to 398.7 HV. When the annealing temperature is increased from 2250 °C to 2300 °C, the hardness of tungsten alloy decreases from 398.7 HV to 387.9 HV.

In summary, with increasing annealing temperature, the hardness of tungsten alloy gradually decreases. After annealing, the tungsten alloy undergoes recrystallization and the grains grow, which reduces the strengthening effect of grain refinement. Furthermore, work hardening from prior large deformation is eliminated, dislocation density and structural defects are reduced and the machinability and plasticity of the material are improved [30].

### 3.4. Fracture Morphology of Tungsten Alloy Wire

Figure 14 shows the fracture morphology of tungsten alloy wire in the initial state. The fracture of the initial tungsten alloy wire is relatively flat, exhibiting typical brittle fracture characteristics. Cleavage fracture typically features cleavage steps, river pattern and cleavage fan. Figure 14b reveals river-like patterns on the fracture surface, indicating that cleavage fracture occurred during the wire drawing. Additionally, deep cracks are observed in the cross-section, propagating through the grains to form transgranular fracture, as shown in Figure 14b. This is attributed to the severe plastic deformation during drawing, which disrupts the brittle grain boundary cohesion, leading to the formation of transgranular cracks [31].

Figure 15a–d display the overall fracture morphology of the tungsten alloy fracture, while Figure 15a1–d1 show the corresponding microscopic morphologies. According to Figure 15a,a1, the tungsten alloy fracture type is brittle fracture. Because the characteristic of the cleavage fracture can be clearly observed in the figure, this indicates that cleavage fracture occurs in the alloy during processing [32]. Figure 15b reveals no significant plastic deformation at fracture, with multiple surface and internal cracks propagating radially from pre-existing defects (e.g., pores or inclusions). Figure 15b1 exhibits terrace-like cleavage steps, confirming brittle fractures. From Figure 15c further demonstrates a brittle, fractured surface without plastic deformation, with Figure 15c1 showing prominent cleavage characteristics. In contrast, Figure 15d displays obvious necking, indicating room-temperature plastic deformation and material ductility [33]. However, Figure 15d1 reveals quasi-cleavage facets with localized dimples, suggesting a mixed ductile–brittle fracture mode [34].

As the deformation degree increases, grains are fragmented into smaller subgrains under complex slip and grain boundary constraints, reducing the average grain size. These subgrains, typically 0.01–10 μm in size, form a subgrain structure confined within original grain boundaries. For most metals, grain refinement induces strengthening according to the classical Hall–Petch relationship. This explains why the hardness of tungsten alloy increases with the decrease in diameter and average grain size under low-temperature processing conditions with deformation ranging from 34.6% to 82.7%. However, when the grain size reaches nanometer level, an inverse Hall–Petch effect occurs where further grain reduction leads to softening [35]. This explains the necking observed on the fracture surface of tungsten alloy samples with 88.4% deformation.

In summary, when the deformation strain increases from 46.4% to 81.2%, the average grain size of tungsten alloy decreases while the hardness increases, so there is no obvious plastic deformation on its fracture surface. When the deformation strain reaches 88.4%, the anti-Hall–Petch phenomenon occurs due to the decrease in average grain size exceeding the critical value. The plastic deformation of tungsten alloy samples becomes obvious, and there is a noticeable necking phenomenon at the fracture surface. This indicates that tungsten alloy wire undergoes a brittle ductile transition behavior with increasing strain.

## 4. Conclusions

(1)The tungsten alloy with 85% rolling deformation exhibits a relatively uniform strength distribution and no significant texture. After annealing, tungsten alloys undergo recrystallization and grain growth. When drawing to ø0.8 mm, the final wire develops a strong <110> filamentary fiber texture.(2)In the process of rotary forging grain growth occurs at high temperatures (1480–1380 °C) along with recrystallization, which increases thermal embrittlement and causes intragranular cracks. Therefore, forging should be performed at lower temperatures (1330–1400 °C). After large-strain processing, subgrain formation and microstructural heterogeneity necessitate annealing prior to further processing.(3)With increasing annealing temperature, recrystallization and grain growth reduce the hardness of tungsten alloy. For the φ8.9 mm specimen, hardness decreases from 466.5 HV to 395.3 HV as temperature rises from 2200 °C to 2300 °C, while the φ5.2 mm specimen shows a decrease from 435.2 HV to 384.9 HV under the same conditions.(4)During wire drawing, increasing deformation strain promotes uniform elongated fibrous structures in the longitudinal cross-section and gradual hardness elevation. At low strains, room-temperature brittle fracture with cleavage features predominates. At an 88.4% strain, the fracture mode transitions to ductile–brittle mixed characteristics.

## Figures and Tables

**Figure 1 micromachines-16-00922-f001:**
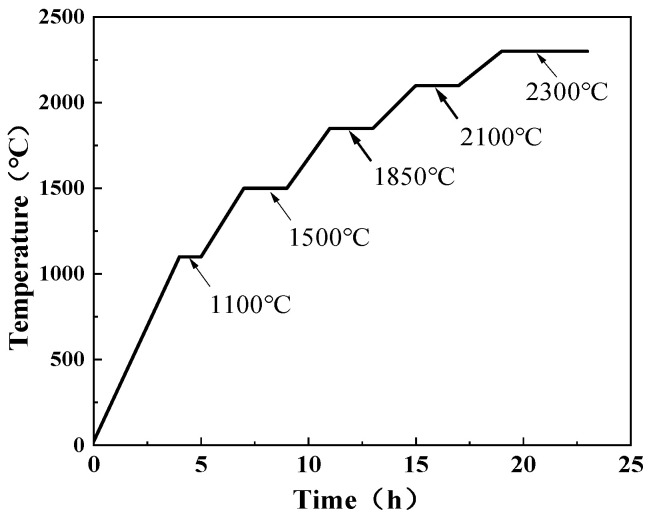
Temperature rise curve of tungsten alloy sintering temperature.

**Figure 2 micromachines-16-00922-f002:**
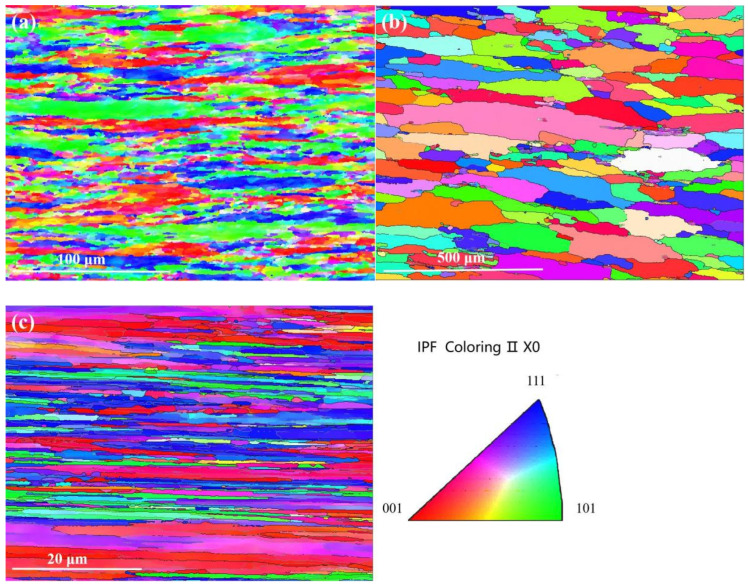
**Figure 2.** Grain boundaries and orientation of tungsten alloy in different states: (**a**) ø8.9 mm 85% rolled deformation; (**b**) ø5.2 mm rod (annealed at 2300 °C) and (**c**) ø0.8 mm wire.

**Figure 3 micromachines-16-00922-f003:**
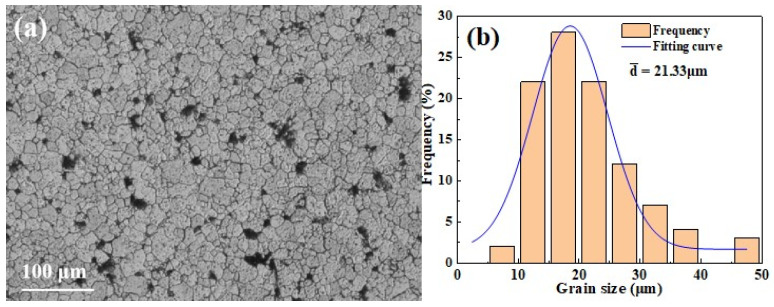
Metallography and grain size distribution of cross-section of 23 mm tungsten alloy rod: (**a**) metallography of tungsten alloy cross-section; (**b**) grain size distribution histogram.

**Figure 4 micromachines-16-00922-f004:**
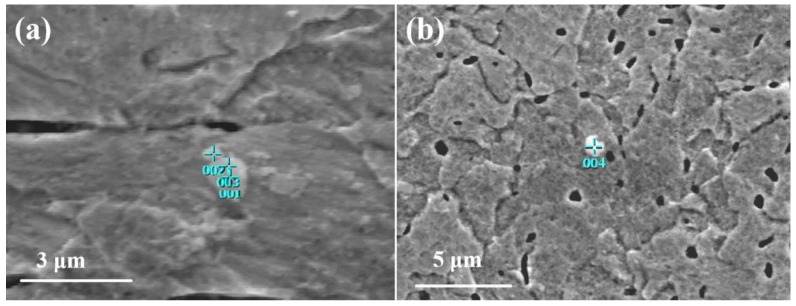
EDS point analysis diagram of white particles. (**a**) Test positions 1-3; (**b**) Test position 4.

**Figure 5 micromachines-16-00922-f005:**
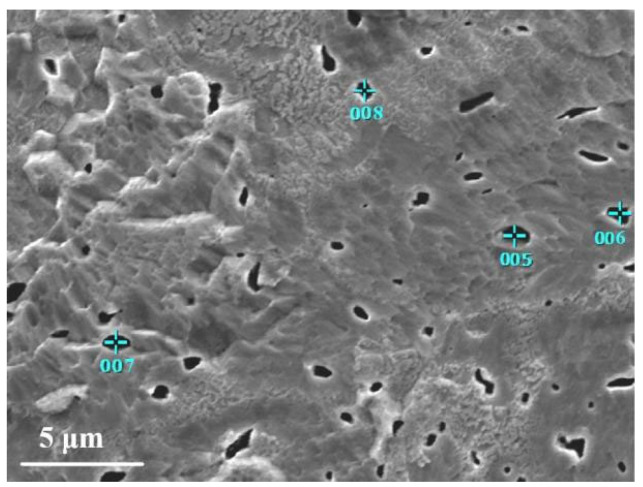
EDS point analysis diagram of black particles.

**Figure 6 micromachines-16-00922-f006:**
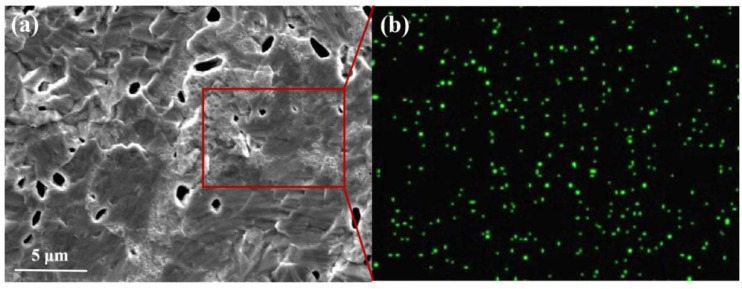
La element surface analysis diagram of ø8.9 mm tungsten alloy rod. (**a**) Micro morphology of doped particles; (**b**) La element distribution status.

**Figure 7 micromachines-16-00922-f007:**
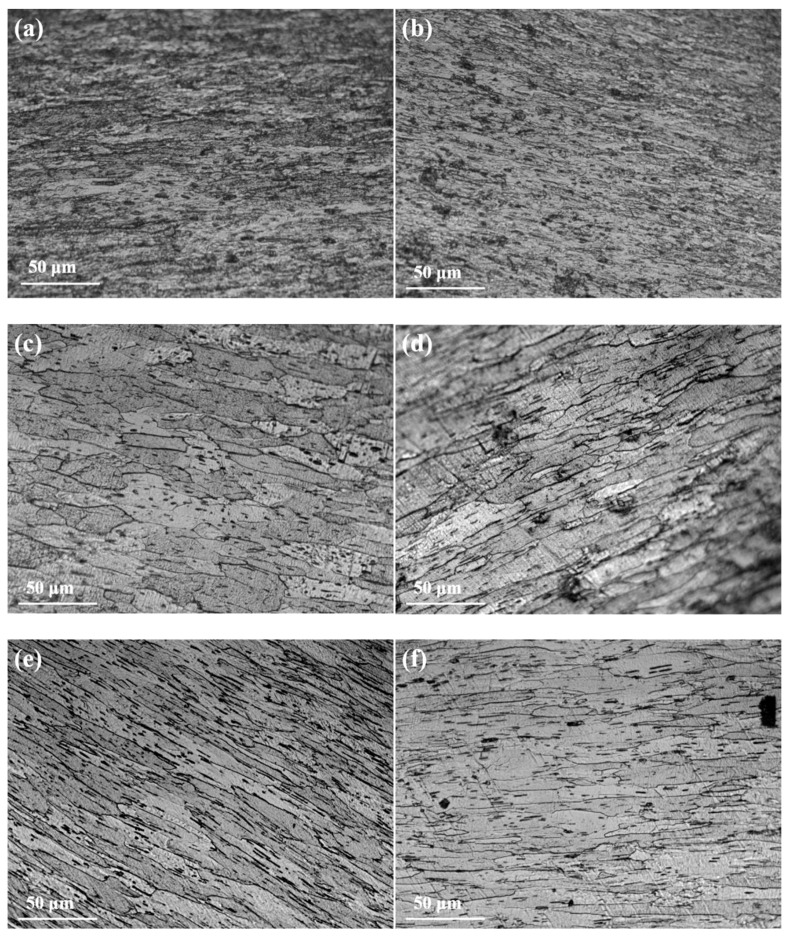
Microstructure of longitudinal section of tungsten alloy: (**a**) 1400 °C, 85%; (**b**) 1480 °C, 85%; (**c**) 1380 °C, 34.6%; (**d**) 1450 °C, 34.6%; (**e**) 1350 °C, 65.9%; (**f**) 1400 °C, 65.9%; (**g**) 1330 °C, 82.7%; (**h**) 1380 °C, 82.7%.

**Figure 8 micromachines-16-00922-f008:**
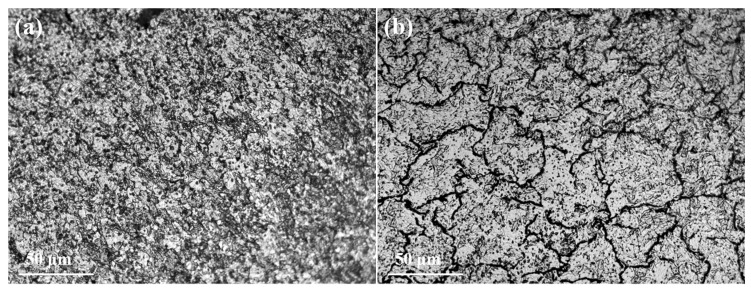
Cross-section microstructure of tungsten alloy: (**a**) 1400 °C, 85%; (**b**) 1480 °C, 85%; (**c**) 1380 °C, 34.6%; (**d**) 1450 °C, 34.6%; (**e**) 1350 °C, 65.9%; (**f**) 1400 °C, 65.9%; (**g**) 1330 °C, 82.7%; (**h**) 1380 °C, 82.7%.

**Figure 9 micromachines-16-00922-f009:**
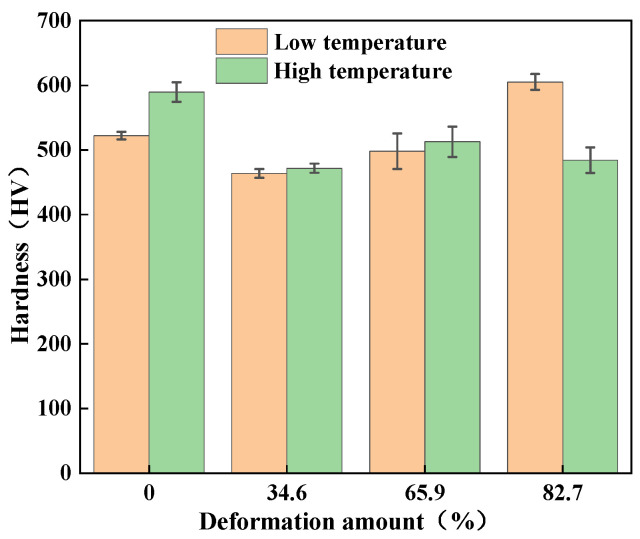
Hardness change diagram of tungsten alloy under different deformation conditions.

**Figure 10 micromachines-16-00922-f010:**
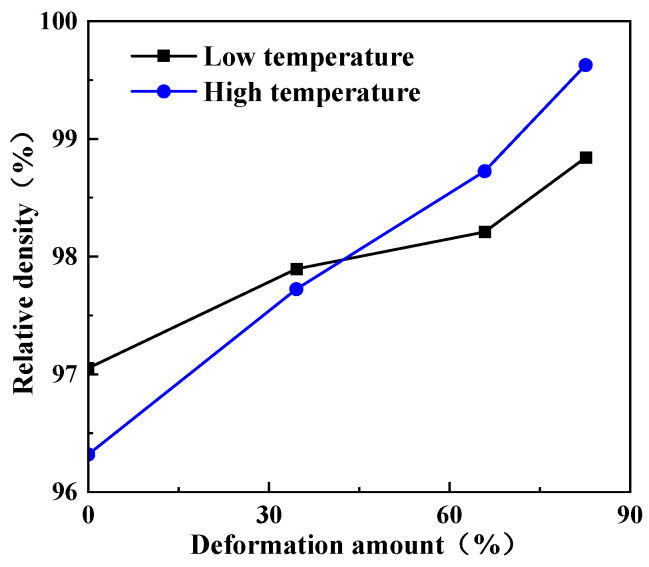
Variation curve of relative density of tungsten alloy samples with deformation.

**Figure 11 micromachines-16-00922-f011:**
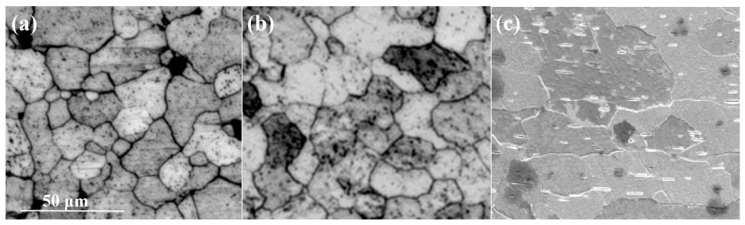
Microstructure and morphology of cross-section of ø8.9 mm tungsten alloy at different annealing temperatures ((**a**) 2000 °C; (**b**) 2100 °C; (**c**) 2300 °C).

**Figure 12 micromachines-16-00922-f012:**
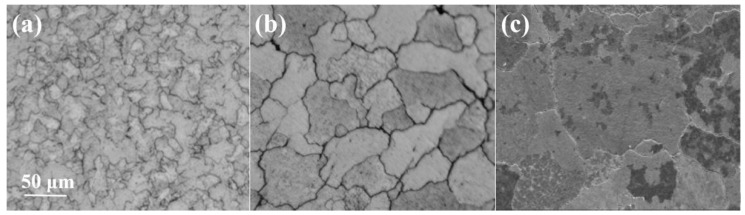
Microstructure and morphology of ø5.2 mm tungsten alloy at different annealing temperatures ((**a**) 2200 °C; (**b**) 2250 °C; (**c**) 2300 °C).

**Figure 13 micromachines-16-00922-f013:**
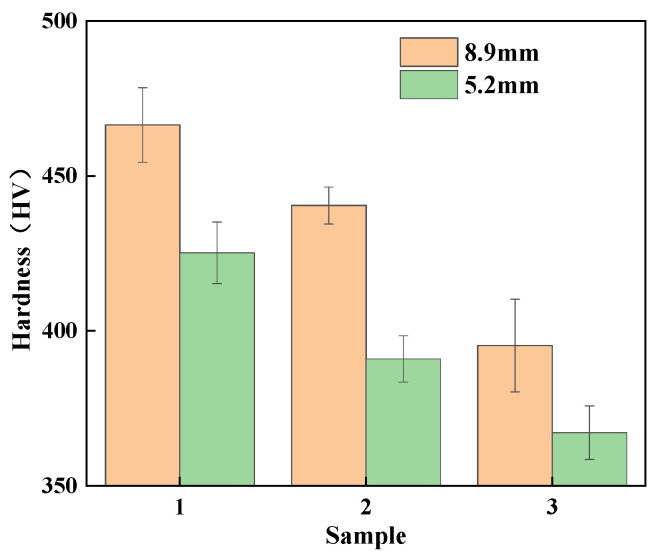
Variation in the Vickers hardness of tungsten alloy at different annealing temperatures.

**Figure 14 micromachines-16-00922-f014:**
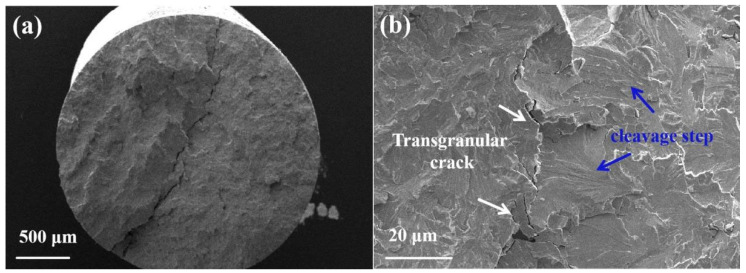
Fracture morphology of initial ø2.35 mm tungsten alloy wire: (**a**) overall morphology; (**b**) micro morphology.

**Figure 15 micromachines-16-00922-f015:**
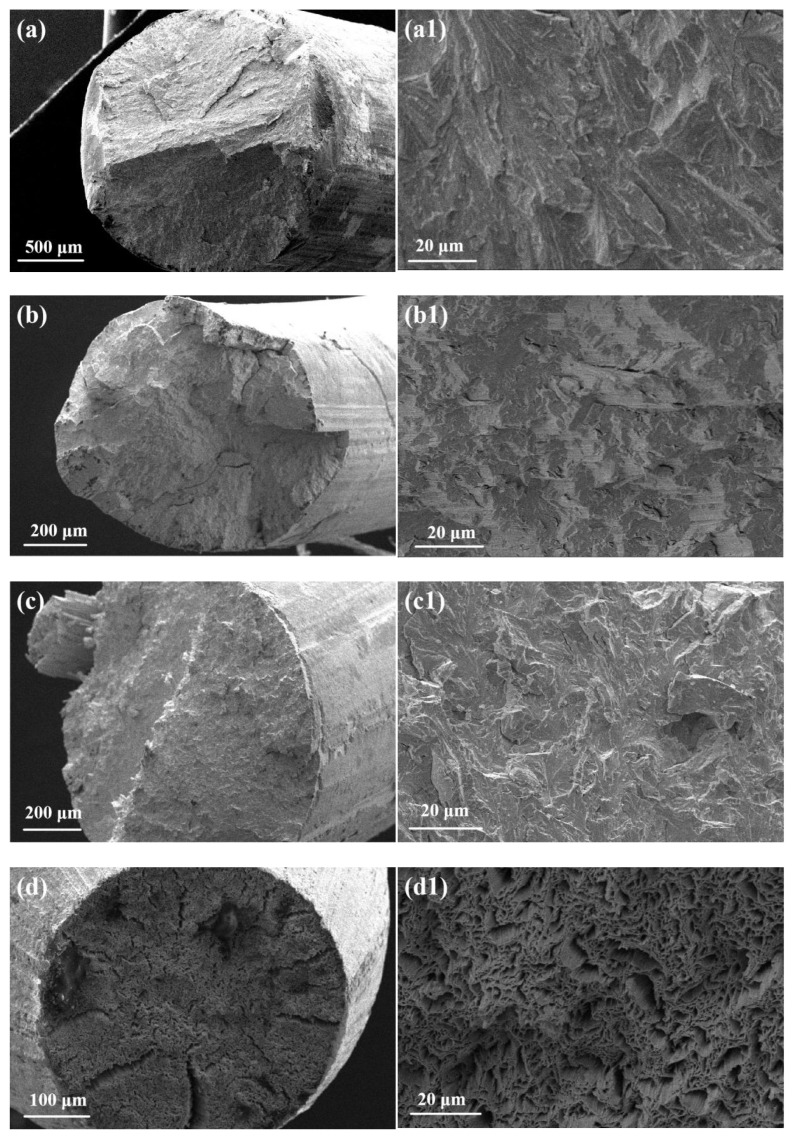
Fracture morphology of tungsten alloy wire under different deformation: (**a**,**a1**) 46.4%; (**b**,**b1**) 68.4%; (**c**,**c1**) 81.2%; (**d**,**d1**) 88.4%.

**Table 1 micromachines-16-00922-t001:** Rotary forging test.

Scheme	Rod Size Change (mm)	Temperature (°C)	Deformation (%)
Low temperature	23→8.9	1400	85
	8.9→7.2	1380	34.6
	8.9→7.2→5.2	1350	65.9
	8.9→7.2→5.2→3.7	1330	82.7
	23→8.9	1480	85
High temperature	8.9→7.2	1450	34.6
	8.9→7.2→5.2	1400	65.9
	8.9→7.2→5.2→3.7	1380	82.7

**Table 2 micromachines-16-00922-t002:** Annealing test.

Scheme	Sample Number	Annealing Temperature (°C)
	1	2000
ø8.9 mm	2	2100
	3	2300
	1	2200
ø5.2 mm	2	2250
	3	2300

**Table 3 micromachines-16-00922-t003:** Drawing test of tungsten alloy wire.

Sample Number	Variation in Wire Diameter (mm)	Deformation (%)
1	2.35	0
2	2.35→1.72	46.4
3	2.35→1.72→1.32	68.4
4	2.35→1.72→1.32→1.02	81.2
5	2.35→1.72→1.32→1.02→0.8	88.4

**Table 4 micromachines-16-00922-t004:** Contents of elements in point analysis of white particles.

	Element	1	2	3	4
Weight percentage (%)	La	1.84	0.70	1.95	1.93
	O	7.88	7.34	6.21	8.96
	W	90.28	91.96	91.84	89.11
Atomic number percentage (%)	La	1.33	0.52	1.56	1.31
	O	49.41	47.59	43.03	52.91
	W	49.26	51.89	55.41	45.78

**Table 5 micromachines-16-00922-t005:** Contents of elements in point analysis of black particles.

	Element	5	6	7	8
Weight percentage (%)	La	19.31	4.59	3.30	3.67
	O	6.20	2.77	5.83	0
	W	74.49	92.64	90.87	96.33
Atomic number percentage (%)	La	14.93	4.65	2.70	4.80
	O	41.57	24.40	41.30	0
	W	43.50	70.95	56.01	95.20

**Table 6 micromachines-16-00922-t006:** Hardness, density and relative density of all samples.

Scheme	Temperature (°C)	Deformation Amount (%)	Vickers Hardness (HV)	Density (g/cm^3^)	Relative Density (%)
Low temperature	1400	0	522.2	18.42	97
	1380	34.6	463.8	18.58	97.9
	1350	65.9	498.0	18.64	98.2
	1330	82.7	605.2	18.76	98.8
High temperature	1480	0	539.2	18.48	97.4
	1450	34.6	431.3	18.62	98.1
	1400	65.9	468.8	18.72	98.6
	1380	82.7	442.7	18.81	99.1

**Table 7 micromachines-16-00922-t007:** Grain size and hardness of annealed tungsten alloy.

	Sample Number	Annealing Temperature (°C)	Grain Granularity (pieces/mm^2^)	Hardness (HV)
	1	2000	1461	466.5
ø8.9 mm	2	2100	1484	440.5
	3	2300	2296	395.3
	1	2200	1496	435.2
ø5.2 mm	2	2250	619	398.7
	3	2300	2263	384.9

## Data Availability

The data used to support the findings of this study are included within the article.

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
