# Peer review of "Study on the Influence of Deformation and Temperature on the Properties of High-Strength Tungsten Alloy Wire"

_micromachines, 2025, doi:10.3390/mi16080922_

Round 1
Reviewer 1 Report
Comments and Suggestions for Authors
This paper studies the effect of deformation and temperature on the microstructure and properties of high-strength tungsten alloy wire. The research focus is interesting and meaningful. However, some points may be addressed before publication, especially the structure of the paper:
- Table 1: How did you determine the temperatures? Why not using the same temperature for low temperature and high temperature tests?
- Section 3 "Results" are not well organised. It is recommended to have clearer focus and title for each subsection. For example: 3.1 Rotary forging; 3.2 Annealing; 3.3 Drawing (this is just recommendation. The authors could have their own preference)
- Lines 121-122: "The small-angle grain boundaries in Figure 2(a) and (c) account for 121 48% and 49%, respectively". Why is the difference so mall under two different states?
- Figure 8: The graphs in the figures are not clear enough.
- Lines 373-374: "As the deformation degree increases, the grain can be crushed and split into many small grains under the action of complex slip and grain boundary constraint". It would be useful to give the average grain size to support this statement. Why did the author give only fracture morphology? This subsection is relatively weaker than other subsections.
Author Response
- Table 1: How did you determine the temperatures? Why not using the same temperature for low temperature and high temperature tests?
Low temperature and high temperature are the differences between two different processing temperatures under the same deformation conditions. As the diameter of the rod decreases, the temperature also needs to gradually decrease to prevent a decrease in material strength due to the growth of grain structure and to avoid the occurrence of cracks during processing.
- Section 3 "Results" are not well organised. It is recommended to have clearer focus and title for each subsection. For example: 3.1 Rotary forging; 3.2 Annealing; 3.3 Drawing (this is just recommendation. The authors could have their own preference)
All have been modified to more appropriate titles.
- Lines 121-122: "The small-angle grain boundaries in Figure 2(a) and (c) account for 121 48% and 49%, respectively". Why is the difference so mall under two different states?
Because Fig. 2a and C are samples processed by rolling and drawing respectively. Both of them are processed by large deformation, although the changes of grain morphology are different, they will lead to the increase of small angle grain boundary, and the similar proportion is just a coincidence.
- Figure 8: The graphs in the figures are not clear enough.
The blurry image has been replaced with a clear one.
- Lines 373-374: "As the deformation degree increases, the grain can be crushed and split into many small grains under the action of complex slip and grain boundary constraint". It would be useful to give the average grain size to support this statement. Why did the author give only fracture morphology? This subsection is relatively weaker than other subsections.
Because the grain size at the fracture cannot be effectively counted at present, only qualitative analysis can be made. The fracture morphology can also reflect the mechanical properties of the material. In addition, some new contents are added in this chapter.

Reviewer 2 Report
Comments and Suggestions for Authors
Manuscript is written in a strange way. There are strange terms, introduction is written in not a clear way, sighinicance of studies was not shown, part of references is not mentioned within the text. I recommend reject and resubmit after significant rewriting.
References are provided in a strange way. It is better to use traditional [] symbols or superscripts. Also, It is most wide-spread, when all the references appear one by one, however within the manuiscript authors cite #6 and then #14, etc. It is a bit confusing. I recommend authors carefully work out with the list of references and references' numeration.
Table 1. Table is a bit confusing. It took couple minutes to uderstand, that there is a strict boundary between "low-temperature" and "high-temperature" schemes, however it is still unclear, why low- and high-temperature schemes were performed under the same temperatures?
Units of hardness (Vickers, Rockwell, Brinnell, etc.) are not mentioned within Materials and Methods part. This should be added.
Authors are recommended carefully check the figures numeration within the text - last paragraph in page 7 provides description of fig. 7, however, authors write about fig. 8 in line 190.
Authors do not show the statistical significance of obtained results. The only exception is fig. 9 and 13. This is better to be corrected.
Authors did not manage to show the significance of their studies. This should be corrected, while it is one of the most important points of all the published articles.
Comments on the Quality of English LanguageAuthors use a bit strange terms. This is better to be corrected.
Also, there are misprints.
Author Response
Manuscript is written in a strange way. There are strange terms, introduction is written in not a clear way, sighinicance of studies was not shown, part of references is not mentioned within the text. I recommend reject and resubmit after significant rewriting.
The whole paper has been revised on a large scale. The academic terms have been standardized and new contents have been added.
References are provided in a strange way. It is better to use traditional [] symbols or superscripts. Also, It is most wide-spread, when all the references appear one by one, however within the manuiscript authors cite #6 and then #14, etc. It is a bit confusing. I recommend authors carefully work out with the list of references and references' numeration.
References have been reorganized and changed to superscript format.
- Table 1. Table is a bit confusing. It took couple minutes to uderstand, that there is a strict boundary between "low-temperature" and "high-temperature" schemes, however it is still unclear, why low- and high-temperature schemes were performed under the same temperatures?
There are two different processing temperatures under the same deformation. "Low temperature and high temperature" is only to distinguish the two processing schemes.
- Units of hardness (Vickers, Rockwell, Brinnell, etc.) are not mentioned within Materials and Methods part. This should be added.
The unit of hardness have been added.
- Authors are recommended carefully check the figures numeration within the text - last paragraph in page 7 provides description of fig. 7, however, authors write about fig. 8 in line 190.
The drawing number error has been corrected.
- Authors do not show the statistical significance of obtained results. The only exception is fig. 9 and 13. This is better to be corrected.
Additional information about the data graph has been added.
- Authors did not manage to show the significance of their studies. This should be corrected, while it is one of the most important points of all the published articles.
The research significance has been added in the introduction.
Round 2
Reviewer 1 Report
Comments and Suggestions for Authors
All have been addressed.